# VEGF Detection via Simplified FLISA Using a 3D Microfluidic Disk Platform

**DOI:** 10.3390/bios11080270

**Published:** 2021-08-11

**Authors:** Dong Hee Kang, Na Kyong Kim, Sang-Woo Park, Hyun Wook Kang

**Affiliations:** 1Department of Mechanical Engineering, Chonnam National University, 77 Youngbong-ro, Buk-Gu, Gwangju 61186, Korea; kdh05010@gmail.com (D.H.K.); naky0607@gmail.com (N.K.K.); 2Department of Ophthalmology, Chonnam National University Medical School and Hospital, Baekseo-ro, Dong-Gu, Gwangju 61469, Korea

**Keywords:** fluorescence-linked immunosorbent assay, lab-on-a-disk, vascular endothelial growth factor, 3D microstructure

## Abstract

Fluorescence-linked immunosorbent assay (FLISA) is a commonly used, quantitative technique for detecting biochemical changes based on antigen–antibody binding reactions using a well-plate platform. As the manufacturing technology of microfluidic system evolves, FLISA can be implemented onto microfluidic disk platforms which allows the detection of trace biochemical reactions with high resolutions. Herein, we propose a novel microfluidic system comprising a disk with a three-dimensional incubation chamber, which can reduce the amount of the reagents to 1/10 and the required time for the entire process to less than an hour. The incubation process achieves an antigen–antibody binding reaction as well as the binding of fluorogenic substrates to target proteins. The FLISA protocol in the 3D incubation chamber necessitates performing the antibody-conjugated microbeads’ movement during each step in order to ensure sufficient binding reactions. Vascular endothelial growth factor as concentration with ng mL^−1^ is detected sequentially using a benchtop process employing this 3D microfluidic disk. The 3D microfluidic disk works without requiring manual intervention or additional procedures for liquid control. During the incubation process, microbead movement is controlled by centrifugal force from the rotating disk and the sedimentation by gravitational force at the tilted floor of the chamber.

## 1. Introduction

As a consequence of recent technological and medical developments, the human average life expectancy has increased by interventions which have focused on age-related disabilities. Analysis of blood biomarkers is essential, which reveals the specificity of biological aging of individuals, such as immune aging, physical function, and anabolism [1]. Among the biomarkers, vascular endothelial growth factor (VEGF) is a signaling molecule to promote the formation of new vessel branches within tumors and progression and metastasis [2]. In particular, VEGF is a pathognomonic biomarker candidate for the diagnosis criteria of age-related macular degradation incidence, which is most related to ischemic eye disease found in the vitreous and aqueous humor in proportion to the VEGF concentration. Observing the variation of VEGF is essential for predicting the effectiveness of therapy and eye disorder prognoses [3,4,5,6]. Research works reported that the increase in the prevalence rate of ocular diseases is correlated with concentrations of VEGF [7,8,9,10,11,12]. Several ophthalmic disorders associated with VEGF concentration, such as pre-proliferative retinopathy [13], ocular ischemic syndrome [14], and retinal vein occlusions [15], can cause retinal ischemia, which can result in irreversible changes in the ocular structures, such as function and anatomy. However, the low concentration of VEGF and limited aqueous humor sample collection make it difficult in clinical practice to measure the VEGF concentration variation.

Fluorescence-linked immunosorbent assay (FLISA) is a plate-based assay technique for quantifying proteins (antibody and antigen), even in the picogram range per milliliter [16,17,18,19]. The protocol using the well-plate platform for FLISA requires the aqueous humor sample to be more than 100 μL in order to detect protein with pg mL^−1^ levels. The amount of the sample that can be extracted for the reliable quantitative analysis is limited to 50–100 μL considering the change in intraocular pressure [20,21]. Moreover, in general, two or more incubation steps are required for 1–2 h each for the binding reaction between proteins in the FLISA protocol [22,23,24,25]. Furthermore, repeating washing steps between incubations is necessary to remove remaining reagents in a liquid. The FLISA protocol is a complicated assay procedure and time-consuming. Consequently, innovation in diagnostic testing uses only small samples and enables high-precision measurements [26,27,28,29,30,31,32].

In several studies focusing on an FLISA using a lab-on-a-disk platform detecting bio-chemicals, the time requirements for the assay protocol could be notably reduced to under 1 h by using microbeads. Lee et al. reported a fully automated immunoassay on a disk platform using whole blood; this entire process was terminated within 30 min through a fully automated disk for infectious disease (antibody of Hepatitis B) detection [31]. Walsh et al. showed that fluorescence intensity with a 1 ng mL^−1^ resolution could be acquired even though all reagents are simultaneously loaded into the chamber, requiring 10 min in the incubation step. FLISA protocol simplification helps improve the quantitative analysis efficiency [20]. These results show that FLISA using microbeads can decrease the protocol time due to the high specific volume of binding reagents. This is the advantage of the microfluidic platforms; only one-tenth of the volume of the reagent is required compared with traditional plate-based FLISA. Moreover, the washing and detection steps are controlled sequentially under the centrifugal force acting on the disk and microbead sedimentation via density difference, without requiring manual intervention. However, implementation of the FLISA protocol on a microfluidic disk-based platform requires the reduction in the reagent volumes as well as precise control. Furthermore, the microfluidic system is segregated for precise fluid control at each step of FLISA; thereby, it has increased the manufacturing costs and complexity of the system. Poly(methyl methacrylate) (PMMA) sheets are commonly used for manufacturing microfluidic chips at a low cost, since they enable the fabrication of thin and transparent chips [33,34,35,36]. However, processing of the PMMA sheet is improper to fabricate the precise geometry of a microfluidic channel to apply the one-step FLISA protocol. For sufficient mixing in the incubation process, novel valve components are tried in the microfluidic system [31,37]. The wax valve components in the disk would physically isolate each incubation process in the protocol. A fully automated detection system is required for the complicated manufacturing process and increases the unit cost to fabricate a microfluidic disk.

From this point of view, this research suggested a method that employs centrifugal and gravitational forces using a three-dimensional microfluidic disk platform to perform the simplified microbead FLISA protocol. This 3D microfluidic disk using multi-material features a hybrid structure comprising a 3D-printed chamber block and laser-cut PMMA layers. The 3D-printed block contains an incubation chamber with a tilted floor for controlling the mixing of reagents through the cyclic movement of microbeads. This microfluidic disk performs precise microfluidic control in the incubation chamber. Furthermore, the washing and detecting steps are achieved without requiring any manual intervention or additional processes. Therefore, the proposed method is expected to diminish the overall cost of manufacturing microfluidic disks.

## 2. Materials and Methods

### 2.1. Fabrication of 3D-Printed Block and Post Treatment

The disk layer is designed using CATIA V5 software for 3D modeling. The chamber model is converted to an STL file format required for the 3D printing software. The incubation chamber layer is fabricated via stereolithography apparatus. The 3D printing model (ProJet MJP 2500 Plus, 3D Systems, USA) has the ability to fabricate the plane surface with roughness average (*R*_a_) of less than 0.4 μm, even at an angle of 45° from the printing surface [38]. The incubation chamber has a volume of 25 μL with a tilted floor. The 3D printing employing SLA enables the fabrication of micro-pillar structures with 15 μm surface roughness (see Appendix A). One side of the chamber is connected to the side wall of the block for liquid transport, and the other side of the chamber is connected to a ventilation hole in the top layer of the block; this configuration is demonstrated in Appendix A. After printing out the incubation chamber block, post-treatment is performed to remove the wax supports and to clean surfaces. First, the supporting wax in the chamber is roughly removed under a water vapor environment at atmospheric pressure for 30 min. Afterward, the hot oil is used to immerse the incubation chamber block for 2 h in order to entirely remove the remaining wax. After cooling the block at room temperature for 10 min, ultrasonic cleaning with detergents and water is used to remove the residual oil in the chamber; each of these processes are continued for 10 min. Lastly, the washed block is desiccated for 24 h in a convection oven to remove the moisture. After assembling 3D microfluidic disk, microfluidic components are washed to remove remaining debris by pipetting using phosphate-buffered saline (PBS, pH 7.4, Sigma-Aldrich, St. Louis, MO, USA) with 100 μL containing 0.5 wt% bovine serum albumin (BSA, Sigma-Aldrich) and once distilled water with 100 μL. Thereafter, the 3D microfluidic disk is desiccated in a convection oven for 24 h at room temperature.

### 2.2. Assembly of the 3D Microfluidic Disk

The 3D microfluidic disk is assembled in a layer-by-layer manner as shown in Figure 1a; the laser-cut PMMA disk layers are sequentially stacked along with the 3D-printed block. The washing chamber and microchannel are assembled by adopting pressure-sensitive adhesive (PSA) sheets between the laser-cut PMMA layers. A laser cutting machine (Nova24 Laser Engraver; Thunder Laser, China) was used to process the PSA film-attached PMMA sheets into the disk shape. The laser-cut pattern is aligned through the 5 mm holes on the disc layer to which disk layers are attached sequentially. Then, a press machine is used to apply 4.0 MPa pressure for the PSA layers between the 3D-printed block and the PMMA disk layers at room temperature for 10 min. The PSA sheet prevents the leakage of liquid from the clearance between the 3D-printed block and the PMMA layers. As shown in Figure 1b, the assembled 3D microfluidic disk features an incubation chamber (blue-dyed water) and a washing chamber (yellow-dyed water). Details of the microfluidic components are presented on the right side of Figure 1b. The microfluidic circuit comprises the following components: (1) incubation chamber, (2) washing chamber, (3) detection region for fluorescence signal, (4) vent hole of the incubation chamber, (5) inlet port of the incubation chamber, and (6) inlet port of the washing chamber. The microchannel bridge between the incubation and washing chambers has a rectangular cross-section with 300 ± 20 μm widths. A cross-sectional schematic view of the microfluidic chamber describes the dimensions as shown in Figure 1c. The microchannel has 500 μm height, which is also the thickness of the PMMA disk. The detection region at the end of the washing chamber has a sharp edge geometry. This edge geometry of the washing chamber is intended to aggregate fluorescence-labeled microbeads in order to intensify the emission light signal.

### 2.3. Preparation of VEGF Reagents

During the reagent preparation, antibodies of VEGF to capture (cAb, Human/Primate VEGF Antibody, R&D Systems, Minneapolis, MN, USA) and detect (dAb, Human VEGF 165 Antibody, R&D Systems) are bound to the surface of microbead and fluorescent dye, respectively. An antibody coupling kit is used for binding the cAb onto the 2.8 μm diameter epoxy magnetic bead surface (Dynabeads antibody coupling kit and M-270 Epoxy microbead, Thermo Fisher, Waltham, MA, USA). This binding procedure requires 24 h in accordance with the protocol manual. The cAb concentration on the microbead surfaces is 20 μg mg^−1^ in 1 mL of the solution with PBS containing 1 wt% BSA. Additionally, a fluorescence conjugation kit is used for binding the fluorescent dye with dAb with 1:1 volume ratio (DyLight 488 Conjugation kit, Abcam, Cambridge, UK); these are then left overnight at room temperature in dark condition. Consequently, the dAb-bound fluorescent dye is diluted in PBS to 1 μg mL^−1^ concentration. Serial ten-fold dilution of the standard VEGF antigen (Recombinant Human VEGF 165, R&D Systems) was performed using PBS to obtain from 1000 to 1 ng mL^−1^ concentration. Additionally, pure PBS without VEGF antigen is used for a control solution. A dextran (Mr 15,000~25,000, Sigma-Aldrich) is dissolved with 20 wt% in PBST (0.05 *v*/*v*% Triton-X100 contained PBS, Sigma-Aldrich) for wash buffer at 60 °C on a magnetic hot plate stirrer for 24 h.

### 2.4. Analysis and Detection of Fluorescence

A microscope was customized for fluorescence measurement at detection region in 3D microfluidic disk, which was placed over the spindle motor instantly after finishing rotation of disk. Fluorescence signal is measured in a darkroom for blocking outside light pollution. The 150 W halogen illuminator with optic lens and filters are used for the light source in the microscope. Optical filtration selectively passes only fluorescence wavelengths using excitation filter (average light transmission, *T*_avg_ over 93% in the 473–491 nm wavelength range), dichroic filter (*T*_avg_ > 93% in the 502–950 nm range, and *T*_avg_ < 7% in the 350–488 nm range), and emission filter (*T*_avg_ > 93% in the 506–534 nm range). Images from fluorescence microscopy are analyzed using image processing software (ImageJ 1.8.0). Inside the border of the washing chamber, the fluorescence excitation signal is amplified. Additionally, background noise is removed by high-pass filtration.

## 3. Results and Discussion

### 3.1. Validation of the Simplified Microbead FLISA Protocol

The immunological assay is one of the general analysis techniques using absorbent or fluorescent materials for the antibody-antigen interaction-based protein level quantification. The enzyme-linked immunosorbent assay (ELISA) process is performed on well-plate platforms; it employs a colorimetric reaction of enzymes that trigger color change according to the conjugation ratio with the substrate. In this technique, the light absorption ratio is varied according to the enzyme concentration. However, the light absorption mechanism of the ELISA protocol has limitations in applying to microfluidic systems. The optical spectroscopy light absorption characteristics have a relationship with the concentration of the reagent, which is expressed as the Lambert-Beer law following Equation (1):(1)$$I=I_{0}\cdot 10^{-\varepsilon cl}$$
where *I* and *I*_0_ are intensity of the transmittance and the incident light to the reagent. The *ε*, *c*, and *l* are the molar absorptivity coefficient (cm^2^ mol^−1^), the molar concentration (mol L^−1^), and the optical path length (cm), respectively. In the microfluidic disk platform, the optical path length is geometrically restricted to spectroscopic analysis, which is the occurring degradation of the spectral resolution [39]. For this part, it is pointless to increase the thickness of the disk for improving the detection resolution in a microfluidic platform. Alternatively, the FLISA method is possible for the superposition of fluorescence signals, which improve the limit of detection for quantitative immunoassay even when using small sample volumes [40,41]. The microfluidic disk platforms employing microbead FLISA protocol are especially helpful in detecting the superposition of excitation signals from the aggregated fluorescence-linked microbeads. This “simplified” FLISA protocol facilitates the incubation step and diminishes the required time for the reaction between antibody and antigen. The reagents are loaded to the incubation chamber concurrently, where they are mixed and bound together.

In this study, the reagents are prepared as described in Figure 2. Prior to applying the microfluidic disk platform, the protocol is confirmed using a 96-well black polystyrene microplate. The variation of the incubation time and reagent volume condition is performed to verify the fluorescence signal linearity in the protocol. The cAb-bound microbeads, dAb-bound fluorescent dye, and human VEGF antigens are simultaneously loaded into the microwell with a volume ratio of 1:1:1. During the incubation process, shaking the incubator is used for gentle mixing of reagents with 120 rpm for 2 h at 37 °C. In the reagent loading and initial incubation steps, the cAb-bound microbead has a high probability opportunity to bind with the VEGF antigen due to the 20 times higher concentration compared with the dAbs in the reagents. After the VEGF antigens are bound to the microbead surfaces, other epitopes of the VEGF antigen are exposed to binding with the dAb-bound fluorescence dye in the reagents during the incubation step. In the washing step, unbound reagents are removed by pipetting for washing three times. The fluorescence microscope is used to confirm the amount of fluorescent dye-coupled microbeads, which affect the fluorescence excitation intensity with regard to VEGF concentration.

The schematic in Figure 3a describes the fluorescent dye-coupled microbead surface formation. The binding between the VEGF antigen-antibody is performed by the specificity of avidity-driven interactions. In Figure 3b, the microbeads are photographed with a long wave pass filter (>630 nm). The red image is based on the relatively high spectral reflectance characteristics of the microbead (ferrite oxide) in the red wavelength (>630 nm) to show the position of the microbeads. Figure 3c is a fluorescence microscopic image showing that the fluorescent dye is well bound to the surface of the microbead. The merged image represents that the green fluorescent dye is located in the same position as the microbeads, as shown in Figure 3d.

The simplified microbead FLISA reduces the required reagent volumes and also the incubation time. The fluorescence signal was analyzed under variations in volume of the reagent and incubation time. The green fluorescence intensities are analyzed for VEGF concentrations of 1 μg mL^−1^ and 1 ng mL^−1^ as well as the pure PBS. In the simplified microbead FLISA protocol, 100 μL reagent volume (which is required volume for the traditional FLISA method) is reduced to 10 and 5 μL, while the incubation time of 2 h is reduced to 1 h.

The fluorescence intensities are presented under the concentration of VEGF antigen as shown in Figure 4. When using reagent volumes of 100, 10, and 5 μL for the concentration of 1 μg mL^−1^ VEGF antigen solution, the fluorescence intensities in the arbitrary unit are 48.08, 35.03, and 36.19, respectively. Moreover, for the 10 μL volume of VEGF antigen solution with 1 μg mL^−1^ concentration, the fluorescence excitation signal is reduced from 35.03 to 28.0 upon decreasing the incubation time from 2 to 1 h. Decreasing the reagent volume (from 100 to 10 µL) and the incubation time (from 2 to 1 h) causes reductions of fluorescent intensity of 34.8% and 20.5%, respectively. This shows that the fluorescence intensity is retained linearly, even when using small reagent volumes and reduced incubation periods. These results show possibilities, despite the reduction in reagent volume and incubation time, for analysis via a microfluidic system using image processing for high resolution through fluorescence signal amplification and the background noise elimination. Thus, the simplified microbead FLISA on the 3D microfluidic disk is performed employing a 1 h incubation time and 10 µL reagent volume. The microbead FLISA protocol shows the possibility of a 0.2~3.0 ng mL^−1^ range of VEGF level detecting, which can be a criterion for clinical signs for diagnosis and treatment [42,43].

A flow chart in Figure 5 compares a traditional sandwich ELISA method [44,45] and the microbead assay method in a microfluidic system [46,47,48]. The protocol can diminish the volume of the antigen solution and the required time for the incubation step up to values of 1/10 and 1/5, respectively.

### 3.2. One-Step Simplified Microbead FLISA Using a 3D Microfluidic Disk

The 3D microfluidic disk using simplified microbead FLISA protocol is completed by a one-step process including incubation, washing, and detection steps in sequence. The schematic of Figure 6 describes the cross-section of the microfluidic circuit along the radial direction of the disk in order to highlight the proposed sequential process. First, the 30 μL wash buffer is loaded into the washing chamber through the inlet port. Then, the reagents with a volume of 10 μL in each of the cAb-bound microbeads, VEGF antigen diluted solution, and dAb-bound fluorescent dye are loaded into the incubation chamber using pipettes.

After loading the reagents, the 3D microfluidic disk is placed on a spindle motor, which is located in a dark room to avoid the photobleaching of the fluorescent dye during the incubation process. The inlet ports are blocked using commercial transparent tape to prevent reagent evaporation during the protocol. For the incubation step, disk rotation is regulated to ensure that the reagents mix sufficiently. In the incubation chamber with the tilted floor, microbeads move along the radial direction of the chamber surface, while accelerating at 260 rpm for 10 s. Afterward, when the disk is stopped for 10 s, the microbeads move back to the disk axis direction along the chamber bottom. The microbead is controlled by regulating rotation to enhance the mixing reagents. The angular velocity of the disk affects the bead movement during sedimentation, which can be expressed as Equation (2), referring to Stokes’ law:(2)$$U_{s}=\frac{2}{9}\frac{\left(\rho_{bead}-\rho_{f}\right)}{\mu }aR^{2}$$
where *U_s_* is the sedimentation velocity; *ρ_bead_* and *ρ_f_* are the microbead and fluid densities, respectively; *μ* is the fluid viscosity; *R* is the radius of the microbead; and *a* is the acceleration. Here, for the microbeads in the incubation chamber with a tilted floor, the acceleration is changed to *a∙cos θ − g∙sin θ*, where *θ* and *g* are the slope of the incubation chamber and gravitational acceleration, respectively. The disk angular velocity with respect to time for the cycle is shown in Appendix A, including the microbead position after each step (reagent loading, reagent incubation, and washing). In the incubation chamber, when the disk rotates and then stops for 10 s, the microbeads move by 5.50 ± 2.86 μm along the radial direction at 260 rpm angular velocity and then move back by 3.22 μm, respectively, according to Stokes’ law (Equation (2)). When the disk is rotating, the distance traversed by the microbead depends on its relative radial position to the central axis of the disk. For the stationary state of the disk, the microbeads on the tilted floor of the incubation chamber are only influenced by gravitational force and move toward the central axis, regardless of their position. The incubation step between reagents is achieved through gentle mixing for 1 h. At the end of the incubation process, the disk is accelerated for microbead sedimentation in the washing step to 5 krpm for 10 s and then continued for 1 min. The unbound dAb-fluorescent dye is separated due to the difference in density when the microbeads pass through the wash buffer. During this process, the fluorescent dye-coupled microbeads as well as the unlabeled microbeads settle together at the end of the washing chamber.

In Figure 7a, the images depict the microbeads with and without fluorescent dye-coupled VEGF at the end of the washing chamber. The dashed yellow line indicates the boundary of the washing chamber. In Figure 7b, the results of the fluorescence area ratio, *A*_f_/*A*_b_, is a parameter evaluated by the fluorescence area (*A*_f_) and the microbead aggregation area (*A*_b_). The value of *A*_f_/*A*_b_ increases with the VEGF concentration between 1 μg mL^−1^ and 0 g mL^−1^. The average coefficient of variation of five points is 5.50% in the calibration curve. To summarize the study, the one-step process is successfully performed employing the simplified microbead FLISA protocol. The proposed multi-material disk structure comprises laser-cut PMMA layers and a 3D-printed block. The microfluidic disk contains a washing chamber with microchannels and an incubation chamber. The pressurized PSA films grasp layers together while preventing leakage of reagents during the processes. In the proposed simplified microbead FLISA protocol, the excitation signal intensity of the fluorescence is linearly related to the VEGF concentration, regardless of adjustment of the incubation time and the volume of reagent. In the 3D microfluidic disk, the entire protocol of the simplified microbead FLISA can be terminated within 1 h without additional manual intervention to process. Furthermore, the fluorescence signals can be detected immediately on the disk after completing the rotation. The obtained green fluorescent images are analyzed through the image processing. The calibration curve shows clearly over 10 ng mL^−1^ to show quantitative detection for the VEGF concentration via the fluorescence area ratio, *A*_f_*/A_b_*. Through the fluorescence images of the fluorescent dye-coupled microbeads, the detection limit of the 1 ng mL^−1^ VEGF concentration is enough to distinguish with 0 g mL^−1^. As a result, VEGF detection can be accomplished to a resolution with ng mL^−1^ through the fluorescence area ratio analysis (*A*_f_/*A*_b_) via a three-dimensional microfluidic system employing the one-step simplified microbead FLISA protocol. The detected VEGF level using a 3D microfluidic disk can be a diagnosis for clinical signs of retinal disorders. In detecting VEGF level with a 1 ng mL^−1^ resolution, difficulties of the quantitative analysis for VEGF concentration still remain in the 3D microfluidic system by using rapid antigen testing. Moreover, the reagents can be detected precisely by operating the disk angular velocity and the number of cycles with bi-direction rotation in order to enhance the resolution for target antigens.

## 4. Conclusions

A simplified microbead FLISA protocol using a multi-material-based 3D microfluidic disk is successfully implemented for the low-level VEGF detection. The 3D microfluidic disk consists of the PSA film-attached laser-cut PMMA layers and a 3D-printed block. In the proposed microfluidic disk, only the component for the incubation step requiring precise control is fabricated using a 3D printing method, whereas the remaining components are constituted of the laser-cut PMMA channel in order to reduce the manufacturing cost. the microbeads are utilized as the substrate for the immobilization of antibodies on their surface to apply the simplified microbead FLISA protocol. Even for a 30 μL volume of the reagents, the enlarged specific surface area of the microbeads provides support for the antigen–antibody interactions. Additionally, in the simplified microbead FLISA protocol, the reagent volumes are only required to be 1/10 compared with the commercial detection method. The linearity of green fluorescence signals is shown with respect to VEGF concentrations. The excited fluorescence signals observed during the detection step are superimposed from the aggregated microbeads at the edge of the washing chamber. The fluorescence area ratio, *A*_f_/*A*_b_, with respect to the VEGF concentration could be confirmed as characteristic with a ng mL^−1^ resolution. Therefore, the 3D microfluidic disk platform can be used to detect VEGF on the sequential benchtop process using only passive mixing in a simple clockwise rotation cycle within one hour. Regardless of the target antigen including VEGF, manipulating the conditions of disk acceleration and cycles with bi-directional rotation enhances the physical contact opportunity to an antigen–antibody interaction by microbead movement control. In the future, it can be applied as a low-cost, high-speed diagnostic system in developing countries for virus detection, such as for COVID-19. This is possible by providing a platform for detecting biochemical targets within an hour by utilizing a 3D microfluidic disk and a simple rotor.

## Figures and Tables

**Figure 1 biosensors-11-00270-f001:**
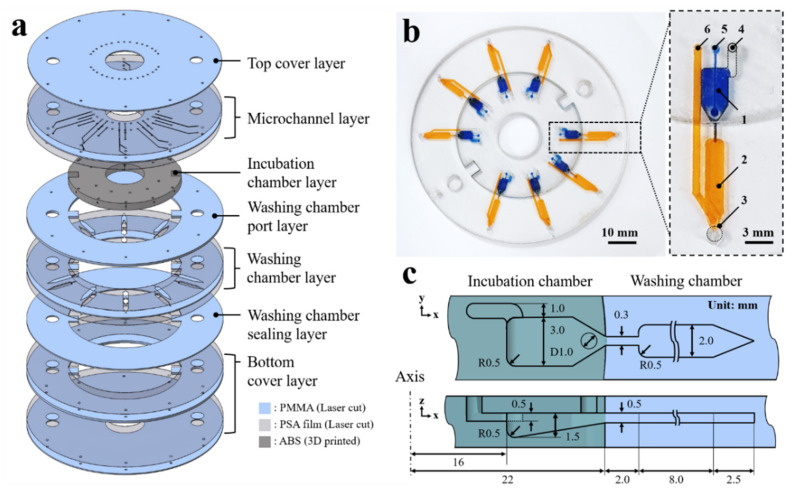
(**a**) Expanded view of the 3D microfluidic disk. (**b**) Image of the 3D microfluidic disk containing dyed water in the incubation (blue dye) and washing chambers (yellow dye). (**c**) Schematic of the cross-sectional illustrating the dimensions of the microfluidic chambers.

**Figure 2 biosensors-11-00270-f002:**
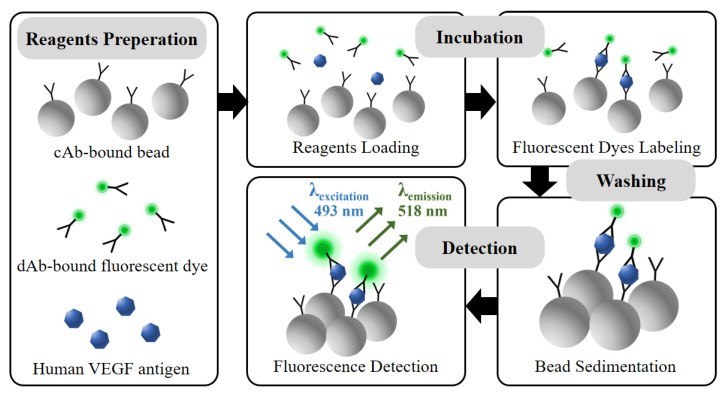
Schematic of the VEGF detection process via simplified microbead FLISA protocol. Biological assay between reagents is performed together in incubation step. The unbound dAb-fluorescent dye is eliminated in washing step. The fluorescence signal is measured for VEGF detection as much as the fluorescent dye attached to the microbead.

**Figure 3 biosensors-11-00270-f003:**
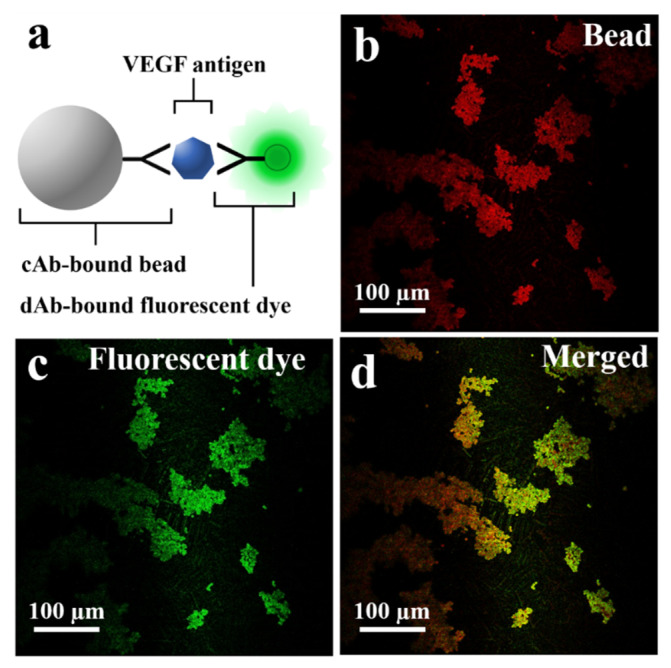
(**a**) Schematic of fluorescent dye-coupled VEGF on the microbeads. (**b**) The microbeads photographed with the longpass filter (>630 nm). (**c**) The dAb-bound fluorescent dye (green fluorescent protein) detected on the surface of the microbeads. (**d**) The merged image for the fluorescent dye and the microbeads.

**Figure 4 biosensors-11-00270-f004:**
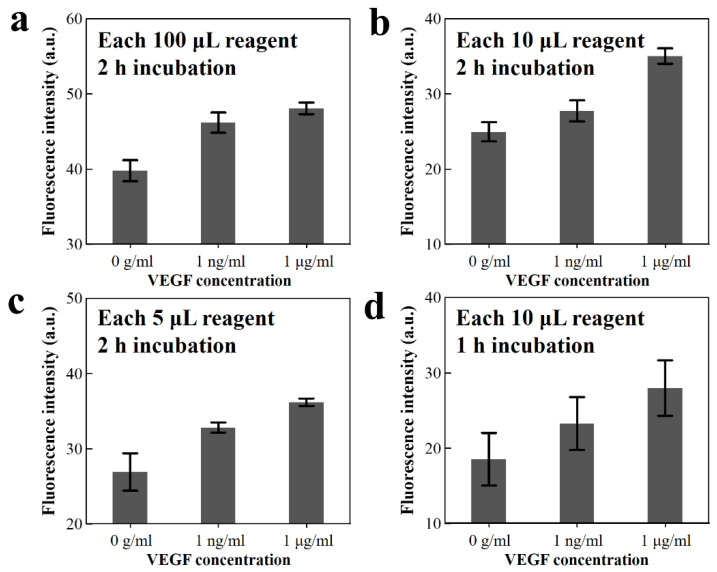
(**a**–**c**) Green fluorescence intensity of the fluorescent dye-coupled VEGF on the microbeads after 2 h incubation, with reagent volumes of 100, 10, and 5 μL, respectively, and (**d**) 10 μL reagent volume for 1 h incubation.

**Figure 5 biosensors-11-00270-f005:**
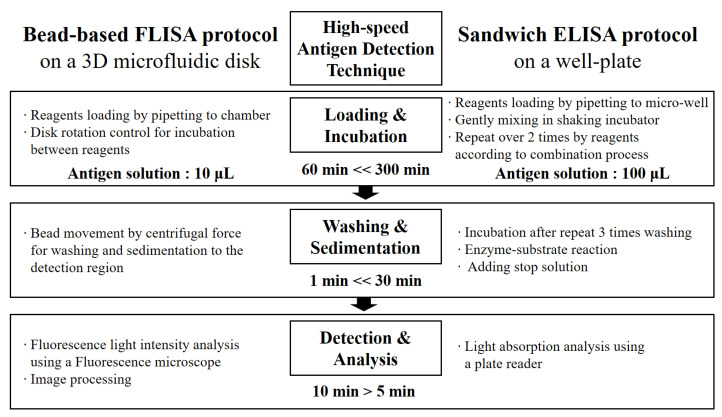
A flow chart comparison of antigen detection techniques between microbead FLISA and traditional sandwich ELISA protocols.

**Figure 6 biosensors-11-00270-f006:**
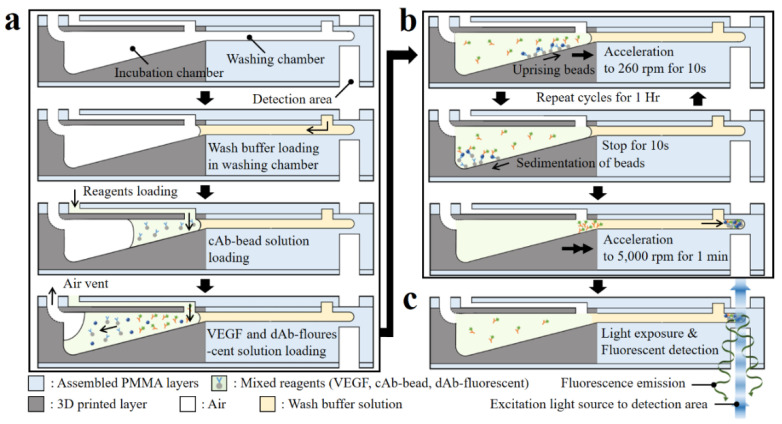
Schematic of sectional view of the microfluidic components highlighting the sequential protocol of the simplified microbead FLISA; the processes indicated are (**a**) loading the wash buffer and reagents, (**b**) incubation, and (**c**) fluorescence detection, respectively.

**Figure 7 biosensors-11-00270-f007:**
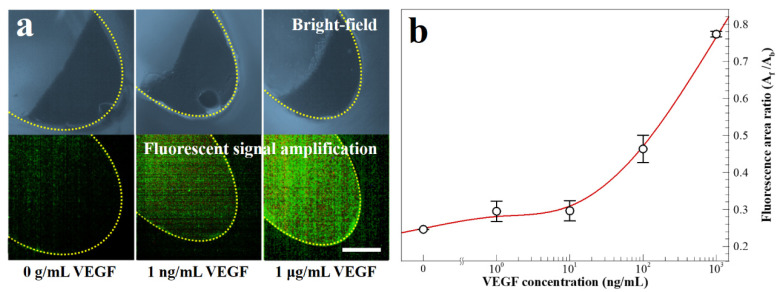
(**a**) Bright-field and fluorescence images for fluorescent dye-coupled microbeads, with VEGF concentrations of 0 g mL^−1^, 1 ng mL^−1^, and 1 μg mL^−1^ (scale bar = 100 μm). (**b**) Fluorescence intensity with varying VEGF concentrations.

## Data Availability

Not applicable.

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
