# Peer review of "VEGF Detection via Simplified FLISA Using a 3D Microfluidic Disk Platform"

_biosensors, 2021, doi:10.3390/bios11080270_

Round 1
Reviewer 1 Report
The article titled “ VEGF Detection via Simplified FLISA using a 3D Microfluidic Disk Platform” presents the design of a microfluidic FLISA plate which is able to detect VEGF with low sample volume and shortened time. This paper validated the one-step FLISA protocol on well plate followed by the evaluation of the effectiveness of detecting VEGF under different concentrations and incubation times. Despite the good experiment design and results presentation, the authors need to address the following comments before the paper is publishable.
- In general, the grammar in the paper needs to be improved. For example, “…is required a more…” in line 70; “…using with…” in line 74; “…enables has possibilities…”; etc.
- The author may need to rewrite the first paragraph of the Introduction. Since the ophthalmic disorder is not mentioned in the title or other parts of the paper, the detection of low concentrated VEGF with small sample volume should be adapted to a broader application, while aqueous humor sample should only be discussed as an example.
- Please add a schematic illustrating the dimensions of the microfluidic chambers.
- Line 201. Please explain why 36ËšC was used instead of 37 Ëš
- Please add scale bars in Fig3 (b-d).
Author Response
Manuscript ID: biosensors-1316193
"VEGF Detection via Simplified FLISA using a 3D Microfluidic Disk Platform" by Dong Hee Kang, Na Kyong Kim, Sang-Woo Park *, Hyun Wook Kang *
We appreciate the referee for her/his helpful and stimulating comments. We have prepared a revised manuscript in accord with the comments of the referee. The revised paragraph, figures and supplementary information have been inserted into the text. Some typographical errors have been corrected. Please see the revised manuscript. As to the specific responses in the revised paper, we would like to note the following modifications.
Reviewer comments:
Comments and Suggestions for Authors
The article titled “VEGF Detection via Simplified FLISA using a 3D Microfluidic Disk Platform” presents the design of a microfluidic FLISA plate which is able to detect VEGF with low sample volume and shortened time. This paper validated the one-step FLISA protocol on well plate followed by the evaluation of the effectiveness of detecting VEGF under different concentrations and incubation times. Despite the good experiment design and results presentation, the authors need to address the following comments before the paper is publishable.
(1) In general, the grammar in the paper needs to be improved. For example, “…is required a more…” in line 70; “…using with…” in line 74; “…enables has possibilities…”; etc.
- In the whole manuscript, the grammar and typos have been corrected. As referee mentioned, the sentences were modified as follows, the first sentence in line 70: "A fully automated detection system is required for the complicated manufacturing process and increases the unit cost to fabricate a microfluidic disk.". Another sentence in line 74: " This 3D microfluidic disk using multi-material features a hybrid structure comprising a 3D-printed chamber block and laser-cut PMMA layers.". And the other sentence in line 91: "The 3D printing employing SLA enables to fabrication of micro-pillar structures with 15 μm surface roughness." respectively.
(2) The author may need to rewrite the first paragraph of the Introduction. Since the ophthalmic disorder is not mentioned in the title or other parts of the paper, the detection of low concentrated VEGF with small sample volume should be adapted to a broader application, while aqueous humor sample should only be discussed as an example.
- As referee mentioned, the VEGF detection is important biomarker to reveal the specific indicator of the biological aging as well as ophthalmic disorder.
- We revised the first paragraph of the Introduction section as below, " In consequence of recent technological and medical developments, the human average life expectancy has increased by the interventions which focused on age-related disabilities. Analysis of blood biomarkers is essential which reveal the specificity of biological aging of individuals such as immune aging, physical function, and anabolism. [1] Among the biomarkers, vascular endothelial growth factor (VEGF) is a signaling molecule to promote the formation of new vessel branch within tumor and progression and metastasis. [2] Especially, VEGF is a pathognomonic bio-marker candidate to diagnosis criteria of age-related macular degradation incidence, which is most related to ischemic eye disease finding in the vitreous and aqueous humor in proportion to the VEGF concentration. The observing of the variation of VEGF is essential for predicting effectiveness of therapy and eye disorder prognoses. [3-6] Research works reported that the increase in the prevalence rate of ocular diseases are correlated with concentrations of VEGF. [7-12] Several ophthalmic disorder associated with the VEGF concentration, such as pre-proliferative retinopathy [13], ocular ischemic syndrome [14], and retinal vein occlusions [15], can cause retinal ischemia, that can result in irreversible changes in the ocular structures as function and anatomy. However, the low concentration of VEGF and limited aqueous humor sample collection make it difficult in clinical practice to measure the VEGF concentration variation."
- And the references, "1. Sebastiani, P.; Thyagarajan, B.; Sun, F.; Schupf, N.; Newman, A. B.; Montano, M.; Perls, T. T. Biomarker signatures of aging. Aging cell 2017, 16, 329-338.", and "2. Longo, R.; Gasparini, G. Challenges for patient selection with VEGF inhibitors. Cancer Chemother. Pharmacol. 2007, 60, 151-170." are added in the manuscript.
(3) Please add a schematic illustrating the dimensions of the microfluidic chambers.
- In the figure 1, the detailed cross-sectional illustration of the microfluidic chambers and its caption are added as below.
- Figure 1: " 1 (a) Expanded view of the 3D microfluidic disk. (b) Image of the 3D microfluidic disk containing dyed water in the incubation (blue dye) and washing chambers (yellow dye). (c) Schematic of the cross-sectional illustrating the dimensions of the microfluidic chambers.".
- Lines 126-127: "A cross-sectional schematic view of the microfluidic chamber describes the dimensions as shown in Fig. 1(c)."
(4) Line 201. Please explain why 36ËšC was used instead of 37 Ëš
- In fact as referee mentioned, the temperature condition during the incubation is 37℃, which was a typo error. The sentence in line 201 was modified as follow, "During the incubation process, shaking incubator is used for gentle mixing of reagents with 120 rpm for 2 h at 37℃.".
(5) Please add scale bars in Fig3 (b-d).
- As referee mentioned, we added scale bar in the Figure 3.

Reviewer 2 Report
In this manuscript, the authors wish to report a simplified microbead FLISA protocol using a multi-material-based 3D microfluidic disk which was successfully implemented for the detection of low-level VEGF. Considering the various advantages of this method over the traditional techniques, the work is important to publish in this journal. Prior to accept for the publication, a minor revision is required which is mentioned below.
Minor comments:
1. What is the source of red fluorescence from the bead in Figure 3b? Although the detection was performed only based on the emission of the fluorophores, authors hardly mentioned about the fluorophores used in this study. Please mention the fluorophores used in the dAb and cAb-bead (if any), along with their excitation and emission collection wavelengths for the respective images in Figure 3b and 3c, as well as for the data in Figure 4 and Figure 7.
2. To demonstrate the high sensitivity of this techniques as claimed by the authors, they should either compare the level of VEGF used in this experiment with that of the real diseased sample or they should perform the VEGF monitoring with a real clinical sample.
Author Response
Manuscript ID: biosensors-1316193
"VEGF Detection via Simplified FLISA using a 3D Microfluidic Disk Platform" by Dong Hee Kang, Na Kyong Kim, Sang-Woo Park *, Hyun Wook Kang *
We appreciate the referee for her/his helpful and stimulating comments. We have prepared a revised manuscript in accord with the comments of the referee. The revised paragraph, figures and supplementary information have been inserted into the text. Some typographical errors have been corrected. Please see the revised manuscript. As to the specific responses in the revised paper, we would like to note the following modifications.
Reviewer comments:
Comments and Suggestions for Authors
In this manuscript, the authors wish to report a simplified microbead FLISA protocol using a multi-material-based 3D microfluidic disk which was successfully implemented for the detection of low-level VEGF. Considering the various advantages of this method over the traditional techniques, the work is important to publish in this journal. Prior to accept for the publication, a minor revision is required which is mentioned below.
Minor comments:
(1) What is the source of red fluorescence from the bead in Figure 3b? Although the detection was performed only based on the emission of the fluorophores, authors hardly mentioned about the fluorophores used in this study. Please mention the fluorophores used in the dAb and cAb-bead (if any), along with their excitation and emission collection wavelengths for the respective images in Figure 3b and 3c, as well as for the data in Figure 4 and Figure 7.
- In the Figure 3b, the microbeads were photographed with long wave pass filter (>630 nm), which is not a fluorophore. The red image is based on the relatively high spectral reflectance characteristics of the microbead(ferrite oxide) in the red wavelength (>630 nm) to show the position of bead. Except for the green fluorescence dye, the other reagents have no opto-chemical properties and not included any fluorophore materials.
In the case of the green fluorescent protein analysis, the excitation and emission collection wavelengths were 493 and 518 nm, where they were used for the green fluorescence images and intensity analysis results in the figures 3(c), 4 and 7.
Therefore, the caption of the figure 3 was modified as follows, "Fig. 3 (a) Schematic of fluorescent dye-coupled VEGF on the microbeads. (b) The microbeads photographed with the longpass filter (>630 nm). (c) The dAb-bound fluorescent dye (green fluorescent protein) detected on the surface of the microbeads (d) The merged image for the fluorescent dye and the microbeads.".
And, in lines 213-218, the paragraph about figure 3 was modified as follows, "The Fig. 3(b), the microbeads are photographed with long wave pass filter (>630 nm). The red image is based on the relatively high spectral reflectance characteristics of the microbead(ferrite oxide) in the red wavelength (>630 nm) to show the position of the microbeads. The Fig. 3(c) is a fluorescence microscopic image showing that the fluorescent dye is well bound to the surface of the microbead. The merged image represents that the green fluorescent dye is located in the same position as the microbeads, as shown in Fig. 3(d).".
(2) To demonstrate the high sensitivity of this techniques as claimed by the authors, they should either compare the level of VEGF used in this experiment with that of the real diseased sample or they should perform the VEGF monitoring with a real clinical sample.
- As shown in the Figure 7, the sensitivity of the simplified FLISA using microfluidic platform is enough to distinguish with 0 g mL-1. Comparing with the real disease samples, the VEGF level is detected in the range of 0.2~3.0 ng mL-1 concentration.
- The detection limit of the 1 ng mL-1 VEGF concentration still can be a criterion for judgement, that is meaningful to catch up a criteria of clinical sign. In the manuscript, the references about the VEGF concentration of real clinical sample from ophthalmic disorder patients, which had been described in lines 242-244, including the references [42, 43] as below.
- Lines 242-244: "The microbead FLISA protocol shows the possibility of a 0.2~3.0 ng mL-1 range of VEGF level detecting, which can be a criterion for clinical signs for diagnosis and treatment. [42, 43]".
- References: "[42] Shimada, H.; Akaza, E.; Yuzawa, M.; Kawashima, M. Concentration gradient of vascular endothelial growth factor in the vitreous of eyes with diabetic macular edema. Invest. Ophthalmol. Vis. Sci. 2009, 50, 2953-2955.", and "[43] Aiello, L. P.; Avery, R. L.; Arrigg, P. G.; Keyt, B. A.; Jampel, H. D.; Shah, S. T.; Pasquale, L. R.; Thieme, H.; Iwamoto, M. A.; Park, J. E.; Nguyen, H. V.; Aiello, L. M.; Ferrara, N.; King, G. L. Vascular endothelial growth factor in ocular fluid of patients with diabetic retinopathy and other retinal disorders. N. Engl. J. Med. 1994, 331, 1480-1487.".
- The mentioned references and our results as above, we also had been discussed in the end of paragraph in the section 3.2 as follows,
- Lines 319-322: "As a result, VEGF detection can be accomplished to resolution with ng mL-1 through the fluorescence area ratio analysis (Af/Ab) via 3-dimensional microfluidic system employing the one-step simplified microbead FLISA protocol. The detecting VEGF level using a 3D microfluidic disk can be diagnosis for clinical signs of retinal disorders."

Round 2
Reviewer 1 Report
The authors have well addressed my questions.